# Association of Proteasome Activity and Pool Heterogeneity with Markers Determining the Molecular Subtypes of Breast Cancer

**DOI:** 10.3390/cancers17010159

**Published:** 2025-01-06

**Authors:** Irina Kondakova, Elena Sereda, Evgeniya Sidenko, Sergey Vtorushin, Valeria Vedernikova, Alexander Burov, Pavel Spirin, Vladimir Prassolov, Timofey Lebedev, Alexey Morozov, Vadim Karpov

**Affiliations:** 1Tomsk National Research Medical Center, Cancer Research Institute, Russian Academy of Sciences, 634009 Tomsk, Russia; kondakova@oncology.tomsk.ru (I.K.); schaschovaee@oncology.tomsk.ru (E.S.); sidenkoevgeniyaaleksandrovna@gmail.com (E.S.); wtorushin@rambler.ru (S.V.); 2Department of Biochemistry and Molecular Biology, Faculty of Medicine and Biology, Siberian State Medical University, 634050 Tomsk, Russia; 3Engelhardt Institute of Molecular Biology, Russian Academy of Sciences, 119991 Moscow, Russia; vedernikova.vo@phystech.edu (V.V.); alexanderburov1998@gmail.com (A.B.); discipline82@mail.ru (P.S.); prassolov45@mail.ru (V.P.); lebedevtd@gmail.com (T.L.); runkel@inbox.ru (A.M.); 4Moscow Center for Advanced Studies, Kulakova 20, 123592 Moscow, Russia; 5Center for Precision Genome Editing and Genetic Technologies for Biomedicine, Engelhardt Institute of Molecular Biology, Russian Academy of Sciences, 119991 Moscow, Russia

**Keywords:** immunoproteasome, chymotrypsin-like activity, caspase-like activity, breast cancer, estrogen receptors, progesterone receptors, Ki67

## Abstract

Breast cancer (BC) heterogeneity determines appropriate drug selection and affects efficacy of treatment. Effective therapy design can benefit from a combination of drugs directed against a specific type of BC with compounds that modulate components of crucial cellular regulatory systems, including the ubiquitin-proteasome system (UPS), which is responsible for the maintenance of protein homeostasis. Here, we revealed mutual influence and a correlation between the proteasome pool heterogeneity and the intratumoral expression of markers determining the BC subtype. Obtained results facilitate the development of novel BC subtype-specific therapy.

## 1. Introduction

Breast cancer (BC) is the most common type of cancer diagnosed in women worldwide [1]. There are four main molecular subtypes that determine the high heterogeneity of BC [2,3]. Molecular subtypes are distinguished based on immunohistochemical determination of the estrogen receptor (ER), progesterone receptor (PR), receptor of epidermal growth factor type 2 (HER-2) and the proliferation marker—Ki-67 expression in the primary tumor. Thus, according to the expression levels of these markers, tumors are divided into luminal A, luminal B, triple-negative and HER-2 positive cancer. In the luminal A subtype, the expression of estrogen and progesterone receptors is high, while the expression of the HER-2 receptor and Ki-67 is low. This molecular subtype occurs in 50–70% of cases, is characterized by a relatively favorable prognosis and is sensitive to hormone therapy [4]. Luminal B subtype is observed in 20–35% of BC patients. It is characterized by high proliferative activity and expression of estrogen and progesterone receptors. The HER-2 receptor can be either overexpressed or not overexpressed. The prognosis for the luminal B cancer is less favorable; both hormone and chemotherapy are recommended for patients with this molecular subtype of the tumor [5,6]. Triple-negative cancers (TNCs) account for 15–20% of newly diagnosed cases of invasive BC and are characterized by the absence of specific therapeutic targets for hormone and targeted therapy. These tumors demonstrate specific patterns of metastasis (usually found in the lungs and brain), aggressive course and poor prognosis for disease-free and overall survival [7,8,9]. HER-2 positive BC is characterized by overexpression of the HER-2 receptor and decreased expression of estrogen and progesterone receptors. This type of tumor is unfavorable in terms of prognosis, as it is associated with aggressive histopathological parameters, such as a high degree of malignancy of the tumor, as well as the presence of metastases to regional lymph nodes. HER-2 positive breast cancer is characterized by the low rates of overall and relapse-free survival in the long-term period, regardless of the size of the primary tumor and engagement of regional lymph nodes [10,11].

The expression of the above mentioned markers is regulated at several levels—transcriptional, translational and posttranslational. The ubiquitin-proteasome system (UPS), involves hundreds of proteins fulfilling the cascade of reactions aimed to specifically tag proteins with ubiquitin and facilitate their degradation. Proteasomes are central elements of the UPS, specifically responsible for the protein hydrolysis. The proteasome is a multisubunit complex, which has three main proteolytic activities: chymotrypsin-like, trypsin-like and caspase-like. Proteasomes perform proteolysis of no longer needed or damaged proteins in the cytoplasm and nucleus, implement conversion of inactive precursor proteins into active ones, generate peptides for presentation by the class I major histocompatibility complex, regulate gene expression and fulfill other important functions. The composition of proteasomes is heterogeneous due to the existence of different catalytic subunits and association of the proteasome core with several different regulators, as well as due to various post-translational modifications [12,13]. The core 20S proteasome consists of four heptameric rings of alpha and beta subunits located one above the other in order α1-7/β1-7/β1-7/α1-7. The N-terminal domains of α-subunits form a gate (pore) that opens or closes the access to the catalytic cavity of proteasome which is formed by the β-subunits. The β subunits are encoded by *PSMB1-10* genes. Three of these subunits perform protein degradation: Proteolytic sites are concentrated at the N-termini of β1 (caspase-like activity), β2 (trypsin-like activity) and β5 (chymotrypsin-like activity) subunits [14]. The constitutive proteolytic subunits β1 (encoded by *PSMB6*), β2 (*PSMB7*) and β5 (*PSMB5*) can be replaced by “immune” analogs LMP-2 (encoded by *PSMB9*), MECL-1 (*PSMB10*) and LMP-7 (*PSMB8*), respectively, during the assembly of so-called immunoproteasomes [15]. These proteasomes are of special interest since they demonstrate altered activity pattern and significantly broaden the repertoire of produced peptides, facilitating antigen presentation [16,17]. The 19S regulatory particles (PA700) could be attached to one or both ends of the 20S proteasome core, resulting in the formation of the 26S proteasome and allowing it to specifically hydrolyze polyubiquitin-conjugated substrates [18]. Except 19S complex, 20S proteasomes are frequently associated with other regulators including 11Sαβ, 11Sγ or PA200. These regulators facilitate ubiquitin-independent hydrolysis of certain proteins and peptides.

The UPS plays an important role in the development of breast cancer [19,20,21]. In particular, proteasomes are involved in the regulation of molecular processes that ensure tumor progression, such as cell proliferation, apoptosis and migration [22,23]. Generally, increased expression of proteasome subunit genes correlates with poor prognosis in BC [24,25]. At the same time, upregulated expression of immunoproteasome subunits is associated with a better prognosis for certain subtypes of BC [26,27,28]. Thus, modulation of the UPS activity and proteasome pool might be considered as a promising strategy for the treatment of the BC.

Currently, BC therapy is based on classic approaches including surgery, chemotherapy, radiotherapy and endocrine therapy, as well as recently introduced options like targeted therapy, immunotherapy and gene therapy [29]. The applicability of the particular therapy approach is dependent on the subtype of the BC. Thus, for example, endocrine therapy is standard for patients with luminal tumors expressing estrogen and progesterone receptors [30], while HER-2 positive breast cancer patients might be treated with drugs targeting specifically HER-2 [31]. Immune checkpoint inhibitors including pembrolizumab and atezolizumab demonstrated efficacy against PD-1-positive triple-negative BC [32,33].

Proteasome inhibitors are long used in cancer therapy. Bortezomib is the first proteasome inhibitor approved by FDA in 2003 for the treatment of multiple myeloma and mantle cell lymphoma. However, bortezomib efficacy against solid tumors including BC is low, and its applicability is limited by severe side effects and resistance development [34]. Although several reports indicated the potential of proteasome inhibitors as a monotherapy against BC [35], combinations of proteasome inhibitors with other drugs likely represent a more promising approach. A recent study reports effective combination of proteasome inhibitor oprozomib (analog of the FDA approved carfilzomib) with doxorubicin for the therapy of triple-negative BC [36]. Moreover, proteasome inhibitors including bortezomib and carfilzomib used in combination with platinum agents or topoisomerase inhibitors exhibit increased efficiency against triple-negative BC cell lines [37]. Interestingly, it has been shown that particular proteasome subunits should be inhibited in order to increase the sensitivity of BC cells to proteasome inhibition [38]. Furthermore, accumulating data indicate that expression of immunoproteasome subunits LMP-2 and LMP-7 correlates with the outcome in patients with triple-negative BC [26,28]. Thus, a recent report indicates that LMP-7-specific inhibitors might be a promising therapeutic option for BC patients at a certain stage of the disease [39].

Together, these data highlight that therapy based on a combination of drugs targeting specific subtypes of BC with rationally selected compounds targeting particular forms of proteasomes might represent a promising therapeutic strategy. Along these lines, we aimed to investigate correlations and possible mutual influence between presence of specific proteasome forms within the proteasome pool and markers determining the subtype of BC.

## 2. Materials and Methods

### 2.1. Patients

The study included 159 patients with primary resectable invasive breast cancer (T1-3N0-2M0). Luminal A cancer was diagnosed in 73 patients (46%), luminal B in 62 (39%), and triple negative molecular subtype cancer was diagnosed in 24 patients (15%). In all breast cancer patients, the diagnosis was verified histopathologically.

Combined treatment of patients began with the surgical stage, including surgical intervention performed in the scope of radical mastectomy or organ-preserving extended surgery. In the adjuvant mode, treatment was prescribed to patients according to indications, taking into account the molecular subtype of the tumor. Adjuvant chemotherapy for luminal A cancer was carried out with a greater prevalence of the process according to the regimen of AC and Taxotere in monotherapy. For luminal B HER-2 negative cancer, chemotherapy was administered in the form of 4–6 courses according to the AC or FAC regimen. For luminal B HER-2 positive cancer, chemotherapy and anti-HER-2 therapy with Herceptin were performed. Patients with triple negative cancer received 6–8 courses of chemotherapy according to the AC + Taxotere regimen in the adjuvant regimen. At the same time, a mandatory component of the systemic treatment of luminal A and luminal B cancer was hormone therapy with tamoxifen or aromatase inhibitors for 5 years. Radiotherapy was performed according to indications. Table 1 shows the distribution of breast cancer patients included in the study. The study was approved by the Local Ethics Committee of the Cancer Research Institute of Tomsk National Research Medical Center (protocol No. 2 from 4 April 2017).

### 2.2. Cell Cultures

The BT-474, SKBR3 and ZR-75 were cultured in RPMI-1640 (Thermo Fisher Scientific, Waltham, MA, USA) medium supplemented with 10% fetal bovine serum (FBS). The MCF-7 cell line was cultured in DMEM (Thermo Fisher Scientific, Waltham, MA, USA) medium supplemented with 10% fetal bovine serum (FBS). Additionally, the growth medium for all cultures included 1 mM sodium pyruvate, 2 mM L-glutamine and antibiotics: streptomycin (100 μg/mL) and penicillin (100 units/mL) obtained from Thermo Fisher Scientific, Waltham, MA, USA. Cells were grown at 37 °C in a humidified atmosphere with 5% CO_2_.

### 2.3. Analysis of Gene Expression and Gene Fitness Data

Data for gene expression in 19,145 tumor samples from 144 datasets were downloaded from R2: Genomics Analysis and Visualization Platform (http://r2.amc.nl, accessed 5 March 2024) and grouped by tumor type into 20 groups as described previously [40]. To calculate the difference between gene expression and other tumor types, we calculated z-scores for mean gene expression for each tumor type. Gene fitness data was downloaded from DepMap data and processed as was described previously [41]. Briefly, dependency scores were calculated for each cell line using DepMap database version 22Q2 as an average gene score from RNAi (Achilles + DRIVE + Marcotte, DEMETER2) [42,43] and CRISPR (DepMap 22Q2 Public + Score, Chronos) [44] screens, and then mean effects for each cancer cell type were calculated. Based on these values for each tumor type, we calculated z-scores representing how different the sensitivity of this cell type is to the sensitivity of other cell types. To compare gene expression in tumors and normal breast tissue, we used data from GSE10780 dataset [45]. To analyze gene expression in different clinical breast cancer subtypes, we used GSE202203 dataset [46]. Single cell expression data for breast tissue were downloaded from CZ CELLxGENE Discover dataset (https://doi.org/10.1101/2023.10.30.563174, accessed 5 March 2024). Single cell data were clustered by Ward D2 algorithm based on weighted gene expression in particular tissue, which was calculated as gene expression multiplied by percentage of cells that express this gene in a cell type.

The statistical tests were chosen based on data distribution as tested by Kolmogorov-Smirnov test. Each gene expression was considered as independent, and comparisons were made between different groups of tumor samples. We used two-sided Student’s *t*-test for comparison of two groups between each other and then applied Benjamini-Hochberg correction for comparison over multiple genes. If there were more than two groups present, we applied one-way ANOVA. The *t*-test and ANOVA were performed in Python using SciPy package. Heatmaps and data clustering were performed in R using ComplexHeatmap package ver. 2.22.0 [47].

### 2.4. Cell Viability Assay

To evaluate the drug sensitivity, BT-474, SKBR3, ZR-75 and MCF-7 cells were seeded in 96-well plates at a density of 2500 cells per well. 24 h later, cells were treated with selective immunoproteasome inhibitor ONX-0914 (Apexbio, Houston, TX, USA) at varying concentrations ranging from 0 to 2 μM and incubated for an additional 72 h. The ONX-0914 was dissolved in DMSO. Therefore, an equivalent concentration of DMSO (0.02%) was used as a control to match the highest drug concentration of 2 μM. Cell viability was assessed using Resazurin Cytotoxicity Assay Kit (Abisense, Sirius, Russia). Supernatant was removed and Resazurin was added to the wells at a ratio of 1:100 in DMEM (MCF-7) or RPMI (BT-474, SKBR3, ZR-75). After 4 h incubation at 37 °C in a 5% CO_2_ atmosphere, absorbance was measured at 570 nm with a reference at 620 nm using a Multiskan FC (Thermo Fisher Scientific, Waltham, MA, USA). The reference signal for each well and the mean signal from wells containing only growth medium and Resazurin were subtracted prior to normalization. The half-maximal inhibitory concentrations (IC50s) were calculated by nonlinear regression with variable slope (four parameters) and robust fitting using the GraphPad Prism software v.8.4.3 (GraphPad Software, San Diego, CA, USA).

### 2.5. Real-Time PCR

Total RNA extraction was performed using Trizol reagent (Invitrogen, Carlsbad, CA, USA). For complementary DNA synthesis, the RevertAid First Strand cDNA Synthesis Kit (Thermo Fisher Scientific, Waltham, MA, USA) was used. Quantitative polymerase chain reaction (qPCR) was performed in triplicate with SYBR Green (Evrogen, Moscow, Russia). The obtained results were analyzed using the CFX96 Touch Real-Time PCR Detection System (Bio-Rad, Hercules, CA, USA). The specific primer sequences were taken from [48]. Cycle threshold (Ct) values were normalized against the endogenous control gene glyceraldehyde-3-phosphate dehydrogenase (GAPDH).

### 2.6. Tumor Sample Preparation

Samples of adjacent and tumor tissues were obtained immediately after radical surgery. When collecting tissue for studying the activity and subunit composition of proteasomes, a thorough analysis of the samples was performed and the unaltered tissue was collected only from areas of epithelial tissue. Adjacent tissue was considered visually and histologically unchanged breast tissue taken at a distance of 1–3 cm from the tumor border. Tissue samples were frozen and stored at −80 °C.

### 2.7. Immunohistochemical Staining

The tissue sections were deparaffinized in xylene, followed by rehydration through a series of graded ethanol solutions (100%, 96%, 80%, 70%) and then washed in distilled water for 5 min. The slides with sections were placed in a plastic holder and immersed in 0.01 M citrate buffer (pH 6.0). Antigen retrieval was performed in a microwave oven in two stages: the first for 7 min at power P = 60, followed by 1 min of cooling with the microwave door open, and then another 7 min at power P = 40 (microwave output power 700 W). After retrieval, the container with the slides was allowed to cool at room temperature for 20 min and then washed in two portions of phosphate buffer for 5 min each. A Peroxidase Blocking Reagent (Dako, Glostrup, Denmark) was applied for 10 min to block endogenous peroxidase, followed by washing in distilled water for 5 min and a final wash in phosphate buffer for 5 min. The antibodies used for analysis included anti-ER (clone 1D5, RTU, mouse), anti-PR (clone PgR636, RTU, mouse, cat. IR068) and anti-oncoprotein c-erbB-2 (HER-2) (working dilution 1:500, rabbit, cat. A0485), all were purchased from Dako (Glostrup, Denmark). To analyze Ki67 expression, mouse clone MM1, RTU antibodies (Leica, Wetzlar, Germany, cat. PA0118) were used. After incubation, the sections were washed in phosphate buffer and treated with secondary biotinylated antibodies. Next, the sections were incubated with the streptavidin-biotin complex for 10 min. After thorough washing in Tris buffer, the sections were treated with a chromogenic substrate (3,3-diaminobenzidine in buffer solution) and incubated with diaminobenzidine for 10 min. A visualization system (Dako EnVision FLEX, Dako, Glostrup, Denmark, cat. K8023) was used. The slides were then counterstained with hematoxylin and mounted in balm. The expression of sex hormone receptors was evaluated using the quantitative Histo-Score method. In this method, both the percentage of positive cells and the expression score were calculated. The expression level of the cell proliferation marker Ki-67 was considered high if more than 20% of cells demonstrated positive nuclear staining. HER-2 expression was considered negative if there was no staining or weak, discontinuous membranous staining. Tissue samples with intense, continuous membranous staining in more than 10% of cells were considered HER-2 positive.

### 2.8. Preparation of Homogenates and SDS-PAGE

Frozen tissue samples (100 mg) were homogenized by freezing/thawing in liquid nitrogen, then resuspended in 300 μL of 50 mM Tris-HCl buffer (pH = 7.5) containing 2 mM ATP, 5 mM MgCl_2_, 1 mM dithiothreitol, 1 mM EDTA and 100 mM NaCl. Samples were centrifuged for 60 min at 10,000× *g* and 4 °C. Cleared homogenates were stored at −80 °C before use.

To characterize the immunoproteasome subunit expression in BC cell lines, cells were washed two times with PBS, collected from the culture plate using a rubber scrapper and lysed in the NP40 lysis buffer (50 mM Tris-HCl pH 8, 150 mM NaCl, 1% NP40, 5 mM EDTA) for 10 min on ice. Samples were centrifuged for 10 min at 10,000× *g* and 4 °C. Cleared homogenates were stored at −80 °C before use.

Electrophoretic separation of proteins for subsequent Western blotting was carried out according to Laemmli [49] in a 13% polyacrylamide gel. Samples were applied in a buffer containing 0.0625 M Tris-HCl (pH 6.8), 2% SDS, 5% 2-mercaptoethanol, 10% glycerol, 0.01% bromophenol blue.

### 2.9. Western Blotting

Following the electrophoresis, polypeptides were transferred to a PVDF membrane (Immobylon, Millipore, Burlington, MA, USA). The content of proteasome subunits was determined by Western blotting. Primary mouse monoclonal antibodies to 20S proteasome subunits α1, α2, α3, α5, α6, α7 (cat. # sc-58412), mouse mAbs to LMP-7 (cat. # sc-100284), rabbit polyclonal antibodies (pAbs) to LMP-2 (cat. # sc-28809), and goat antibodies to the subunit PA28β (cat. # sc-23642) were obtained from Santa Cruz Biotechnology, Dallas, TX, USA. Mouse mAbs to the subunit Rpt6 were purchased from Enzo (Enzo life sciences, Farmingdale, NY, USA cat. #BML-PW9265-0100). Blocking, washing and incubation of membranes with primary antibodies to proteasome subunits (dilution of 1:500), as well as secondary antibodies (goat anti-mouse HRP conjugates (cat. # sc2005) and goat anti-rabbit HRP conjugates (cat. # sc2004) diluted 1:10,000 (both from Santa Cruz Biotechnology, Dallas, TX, USA)) during Western blotting were carried out using the iBind Western Device membrane processing system according to the manufacturer’s protocol (Thermo Fisher Scientific, Waltham, MA, USA). When proteasome subunit expression was assessed in cell lysates, membranes were incubated with rabbit polyclonal antibodies (pAbs) to LMP-2 (cat. # ab-3328), or rabbit pAbs to LMP-7 (cat. # ab-3329), or rabbit pAbs to the MECL-1 (cat. #EPR14902); all antibodies were obtained from Abcam, Cambridge, UK. Goat anti-rabbit HRP conjugates were used as the secondary antibodies (cat. # ab-97200). For the signal normalization, blots were stripped and incubated with antibodies to β-actin (Santa Cruz Biotechnology, Dallas, TX, USA cat. # sc-81178). Optical density of revealed bands was assessed using the ImageJ software ver.1.54i (https://imagej.net/software/fiji, accessed 5 March 2024), and the level of proteasome subunits in unaltered tissue was considered as 100%. The results were expressed as a percentage of the content of the studied proteins in unchanged tissue.

### 2.10. Determination of Proteasome Activities

Chymotrypsin-like (ChTL) and caspase-like (CL) proteasome activities were determined in cleared tissue homogenates of BC patients by hydrolysis of the fluorogenic oligopeptides: Suc-LLVY-AMC or Z-LLE-AMC (Sigma-Aldrich, St. Louis, MO, USA cat. # S6510; C0483), respectively. The reaction mixture contained 20 mM Tris-HCl (pH 7.5), 1 mM dithiothreitol, 30 μM of the corresponding substrate, 5 mM MgCl_2_ and 1 mM ATP. Reactions were carried out at 37 °C for 20 min. The reaction results were recorded using a Hitachi-850 fluorometer (Hitachi, Tokyo, Japan) at an excitation wavelength of 380 nm and emission wavelength of 440 nm. The amount of enzyme that hydrolyzes 1 nmol of Suc-LLVY-AMC or Z-LLE-AMC within 1 min was taken as a unit of proteasome activity. To assess the activity of contaminant proteases, control reactions were performed containing the proteasome inhibitor, MG132 (Sigma-Aldrich, St. Louis, MO, USA cat. # C2211). The specific activity of proteasomes was expressed in units of activity per 1 mg of protein. Protein content was determined using the Lowry method [50].

### 2.11. Statistical Analysis

Statistical analysis was carried out using IBM SPSS Statistics v.20.0 (StatSoft, Tulsa, OK, USA) and Statistica 10.0 (StatSoft, Tulsa, OK, USA) software. The normality of the distribution of the studied samples was evaluated using the Kolmogorov–Smirnov test. The significance of differences between groups was determined by the nonparametric Mann–Whitney and Kruskal–Wallis tests for independent samples. To assess the relationship and influence of parameters, Spearman’s rank correlation coefficient (R) and linear regression method (one-factor analysis) were used.

## 3. Results

### 3.1. Proteasome Gene Expression Differs Between Types of Breast Cancer

First, we compared proteasome beta subunits (*PSMB1-10*) gene expression in breast cancer with other malignant tumors using data for 19,145 patient samples from 144 datasets covering 20 cancer types (R2: Genomics Analysis and Visualization Platform, http://r2.amc.nl, accessed 5 March 2024) and analyzed BC cell lines dependency on gene expression (from DepMap data) using a pipeline published previously [41]. No evident difference in expression of *PSMB1-10* genes in breast tumors compared to other tumor types was revealed; however, we detected that BC cell lines were much more dependent on *PSMB8* expression (Figure 1a,b). Immunoproteasome genes are known to be mainly expressed by immune cells, meaning that expression of *PSMB8* and *PSMB9* genes in tumor samples could be driven by infiltrating immune cells. To analyze how *PSMB1-10* genes are expressed in breast tissue, we analyzed single cell expression data for breast tissues from CZ CELLxGENE Discover dataset (https://doi.org/10.1101/2023.10.30.563174, accessed 5 March 2024). As expected, *PSMB1-7* had uniform expression across all cell types, while *PSMB8* and *PSMB9* were highly expressed in immune cells (Figure 1c). At the same time, significant expression levels of *PSMB8-9* were detected in endothelial cells and fibroblasts. Considering that *PSMB8* knockdown/knockout reduces proliferation of some breast cancer cell lines (Figure 1b), immunoproteasome expression in cancer cells may play an important role in BC carcinogenesis. Along these lines, high *PSMB8* expression was associated with favorable prognosis (https://www.proteinatlas.org/ENSG00000204264-PSMB8/pathology/breast+cancer, accessed 5 March 2024).

Next, we compared *PSMB1-10* gene expression between breast carcinomas (n = 41) and histologically normal breast tissues (n = 143) from the GSE10780 dataset [45]. Expression of most *PSMB* genes was higher in malignant tissues, and there was a significant difference for *PSMB2-5* and *PSMB9* genes (Figure 1d). Since *PSMB* genes encode proteasome beta subunits and most of them were elevated in tumor samples, we analyzed how their co-expression changes in tumors vs. normal tissue. For correlation analysis in breast cancer tumors, we assessed *PSMB1-10* gene expression in 3207 tumor samples from GSE202203 dataset [46]. For normal samples, *PSMB2*, *PSMB3*, *PSMB5*, *PSMB7* and, interestingly, *PSMB10* showed moderate correlation with other subunits (Figure 1e). For tumors in general, correlations for *PSMB1-7* were weaker, meaning that upregulation of particular *PSMB* gene may occur independently from other subunits. However, *PSMB8-10* showed strong correlation between each other, but not with other subunits, indicating that in tumors they are upregulated simultaneously and probably are co-regulated by the same factors, as opposed to normal tissue (Figure 1e,f). To elucidate the role of *PSMB* expression in breast cancer progression, we analyzed gene expression in different clinically annotated breast cancer subtypes from GSE202203 dataset [46].

Expression of all *PSMB* genes was higher in poorly differentiated aggressive tumors with grade 3 according to Nottingham histologic grading system (Figure 2a). The *PSMB* genes expression was significantly higher in ER-negative tumors (Figure 2b), and *PSMB4*, *PSMB9* and *PSMB10* expression was higher in PR-negative tumors (Figure 2c). There was no significant difference between HER-2-positive and HER-2-negative tumors, except for *PSMB5* (Figure 2d), indicating that *PSMB* expression increase in high-grade tumors is likely due to their high expression in hormone-negative tumors. At the same time, expression levels of *PSMB8* and *PSMB9* genes were near significantly elevated in HER-2-positive tumors (Figure 2d). The *PSMB2-5* and *PSMB9* expression was also increased in Ki-67-positive tumors (Figure 2e), which is consistent with increase of these genes’ expression in malignant tumors compared to normal tissue. We also noted increase of *PSMB2-4* and *PSMB8-10* in tumors with metastasis to 3–10 lymph nodes (N1 and N2); however, tumors that have spread to 10 or more lymph nodes (N3) had the same expression of those genes as tumors without involvement of lymph nodes (Figure 2f). Notably, *PSMB9* expression had the highest amplitude of difference for grade 3 tumors, ER-negative, PR-negative and N2 tumors, suggesting the important role of immunoproteasomes in breast oncogenesis. Thus, our results indicate that breast cancer types differ in terms of proteasome subunit expression pattern, and together with strong *PSMB8-10* co-correlation in tumors (Figure 1e), these data suggest exclusive function of immunoproteasomes in biology of the BC.

### 3.2. Immunoproteasome Subunit Expression Is Different in Cell Lines Obtained from Different BC Subtypes

To investigate if the revealed differences in gene expression correlate with the amount of proteasome subunits in cancer cells, we analyzed the relative levels of mRNAs encoding the immunoproteasome subunits *PSMB8*, *PSMB9* and *PSMB10* in four breast cancer cell lines: MCF-7, SKBR3, ZR-75-1 and BT-474. Notably, the HER-2-negative cell line MCF-7 exhibited a dramatic difference, showing an almost complete absence of mRNA encoding *PSMB8* and *PSMB9* compared to the other three HER-2-positive lines (Figure 3a), which was confirmed by lowest levels of corresponding proteins LMP-7 and LMP-2 (Figure 3b,c). In contrast, no significant difference in the expression of *PSMB8* was observed among the HER-2-positive cell lines [51] (Figure 3a). SKBR3 cells exhibited the highest levels of *PSMB8*, *PSMB9* and *PSMB10*. Moreover, SKBR3 and ZR-75-1 demonstrated highest levels of immunoproteasome subunits (Figure 3b,c). To investigate the sensitivity of BC cell lines to immunoproteasome inhibition, the cells were treated with LMP-7 specific inhibitor—ONX-0914 and IC50 values were determined. Interestingly, SKBR3 and ZR-75-1 cells were found to be more sensitive to ONX-0914 (IC50 = 30 nM and IC50 = 122 nM, respectively) when compared to the other cell lines (MCF-7—IC50 = 181 nM; BT-474–IC50 = 303 nM) (Figure 3d). These results indicate that BC cell lines differ by the levels of immunoproteasome expression, and cell lines with high immunoproteasome expression, in general, demonstrate higher sensitivity to immunoproteasome inhibition.

### 3.3. Proteasome Activities Are Increased in Breast Cancer Tissues

To investigate if the revealed differences in gene expression correlate with the amount of proteasome subunits and proteasome activity in tumor tissues, the immunohistochemical staining of 159 samples of tumor and adjacent tissues using antibodies to ER, PR, HER-2 and Ki-67 was performed. The samples were divided into three molecular subtypes of breast cancer: luminal A (positive expression of ER and/or PR, negative HER-2 status, Ki-67 expression < 20%), luminal B (positive expression of ER and/or PR, negative HER-2 status, Ki-67 expression ≥ 20% or positive expression of ER and/or PR, positive HER-2 status at any Ki-67 level) and triple negative molecular subtype of breast cancer (negative expression of ER and PR, negative HER-2 status) (Figure 4a).

Next, we determined the median values of ChTL and CL proteasome activities in breast cancer tissues. It has been shown that chymotrypsin-like activity was 2.4-fold higher (*p* < 0.05, Mann-Whitney), and the caspase-like activity was 2.7-fold higher (*p* < 0.05, Mann-Whitney) in tumors compared to the adjacent tissue (Figure 4b), corroborating revealed increased expression of proteasome genes in BC (Figure 1d). After that, a comparison of proteasome activity was performed between different molecular subtypes of BC (Figure 4c). Surprisingly, no statistically significant differences in proteasome activity between different BC subtypes were revealed (*p* > 0.05). Nevertheless, luminal A cancer was characterized by the lowest values of ChTL and CL proteasome activities. Luminal B subtype was characterized by the highest CL activity of proteasomes; triple-negative tumors were characterized by the highest values of ChTL activity.

### 3.4. The Content of Proteasome Subunits in Breast Cancer Tissues Varies Depending on the Tumour Subtype

Changes in proteasome activity may be associated with the heterogeneity of catalytic subunits within proteasomes and changes within the amount of proteasome regulators. Hence, we studied the amount of 20S proteasome alpha subunits (α1, α2, α3, α5, α6, α7) and immunoproteasome subunits (LMP-2 and LMP-7), as well as the quantity of 19S and 11Sαβ regulator subunits in tumor tissues by Western blotting (Figure 4d). Though significant differences were observed between samples from individual patients (Figure 4d), general analysis of the content of the studied subunits did not show statistically significant differences between BC and adjacent tissues (Figure 4e). Notably, a single protein band was frequently observed in Western blotting with antibodies to α1, α2, α3, α5, α6, α7. This effect might be a result of overlap of signals from proteasome subunits with close MW, post-translational modifications of proteasome subunits or different avidity of monoclonal antibodies directed to particular subunits. To reduce the amount of relevant heterogeneity factors, we compared the expression patterns of proteasome subunits within the individual molecular subtypes of BC (Table 2).

From the bioinformatics data and analysis of BC cell lines, it could be anticipated that luminal A cancer samples will demonstrate the lowest levels of immunoproteasome subunits, whereas triple-negative cancer samples—the highest. Indeed, a statistically significant increase in the level of the immune subunit LMP-7 (encoded by *PSMB8*) was found in HER-2 positive luminal B subtype compared to luminal A, while the LMP-2 (encoded by *PSMB9*) subunit was significantly increased in triple-negative tumors (Table 2). At the same time, in luminal A cancer compared to luminal B cancer, a 1.62-fold increase in the content of α1, α2, α3, α5, α6, α7 subunits was detected (*p* = 0.049). Surprisingly, the content of the PA28β subunit was 2.83-fold higher in luminal A cancer compared with patients with a triple-negative tumor phenotype (*p* = 0.00). Thus, obtained results indicated that proteasome pool composition in BC tissues partially coincided with the revealed patterns of *PSMB1-10* subunit expression. High proteasome activity was also partially associated with the amount of proteasome subunits.

### 3.5. The UPS Activity Correlates with Expression of BC Subtype Markers

To study the participation of the UPS in the BC subtype formation, we estimated the correlation and regression relationships reflecting possible link and influence between estrogen, progesterone receptors, Ki-67 and proteasome activities. The percentages of cells expressing ER, PR, Ki-67, and the presence or absence of the HER-2 receptor overexpression were taken into account. Correlations were identified between the markers that determine the molecular subtype of BC: percent of cells expressing ER and percent of cells expressing the PR positively correlated with each other (R = 0.63; *p* = 0.000); Ki-67 and HER-2 demonstrated a weak positive correlation (R = 0.15; *p* = 0.04); percent of ER and Ki-67 expressing cells revealed a negative correlation (R = −0.39; *p* = 0.000); percent of PR and HER-2 expressing cells showed a weak negative correlation (R = −0.23; *p* = 0.01). Proteasome components also work interconnectedly: In tumor tissue, direct correlations of moderate strength were obtained between ChTL and CL proteasome activity (R = 0.58; *p* = 0.000) (Table 3).

In addition, using the linear regression method, relationships that reflected the possible influence of the percentage of cells expressing estrogen receptors on the ChTL activity of proteasomes (β = 0.74; *p* = 0.000) and on the CL activity of proteasomes (β = 1.89; *p* = 0.000) were identified (Table 4). Obtained result indicated that the content of progesterone receptors can also influence the ChTL activity (β = 0.88; *p* = 0.000) and the CL activity of proteasomes (β = 1.88; *p* = 0.000). Furthermore, the Ki-67 expression level correlated positively with ChTL (β = 1.66; *p* = 0.000) and CL (β = 3.04; *p* = 0.000) activities of proteasomes. It should be noted that proteasome activities also demonstrated a possible effect on the content of ER, PR, Ki-67, but the values of the linear regression coefficients (β) were lower. Thus, the value of proteasome ChTL activity in the tumor influenced the percent of ER (β = 0.46; *p* = 0.000), PR (β = 0.31; *p* = 0.000) and the Ki-67 (β = 0.25; *p* = 0.000) expressing cells. While the level of CL activity can influence the content of ER (β = 0.15; *p* = 0.000), PR (β = 0.09; *p* = 0.000) and the Ki-67 (β = 0.07; *p* = 0.000) in the tumor (Table 4).

## 4. Discussion

It is well established that UPS modulates levels of protein markers that determine subtypes of breast cancer. Estrogens are drivers of ER+ BC. At the same time, contemporary studies suggest their role in ER− breast cancer as well. It has been shown, that UPS regulates the expression of ER in two ways: by stimulating degradation of SE translocation oncoprotein, promoting its disassociation from p53 and PP2A, leading to ER expression stimulation; and by regulating the content of ER via proteasomal degradation [52,53]. However, the role of proteasomes in the regulation of ERα is not limited to the control of protein content, since chymotrypsin-like activity of proteasomes is required to regulate the expression of ERα encoding gene (*ESR1*) [54]. Thus, proteasome inhibitor bortezomib caused a decrease in *ESR1* mRNA levels [54]. The progesterone receptor (PR) modulates estrogen receptors α (ERα) action in BC, regulates the proliferation of mammary gland cells and differentiation of breast tissue [55,56]. Degradation of PRs is also partially mediated by proteasomes. Ligand-dependent reduction in the expression of PRs occurs through phosphorylation of the receptor, its ubiquitination and subsequent degradation within the 26S proteasome [57]. Proteolysis of PR is preceded by phosphorylation at Ser294, which can be triggered through activation of fibroblast growth factor-7/fibroblast growth factor receptor-2, leading to ubiquitination of progesterone receptors [58]. In particular, it was shown that activation of PR by phosphorylation at Ser294 is accompanied by nuclear localization and subsequent degradation of the receptor [59]. Moreover, the activation or inactivation of the HER-2 receptor is dependent on the UPS-mediated turnover. It was demonstrated that proteasome inhibition prevents HER-2 lysosomal degradation via blockade of receptor internalization [60]. The relationship between the ubiquitin-proteasome system and the Ki-67 is inextricably linked with the regulation of proliferative activity [61]. CDK4/CDK6 inhibition decreased Ki-67 expression by G1 arrest and stimulated degradation of Ki-67 in the 26S proteasomes [62,63].

However, currently, little is known about the mutual influence of molecules that determine the molecular subtype of BC and proteasome functional activity. The present study showed regression relationships reflecting the possible mutual influence between estrogen receptors, progesterone receptors, proliferative activity and immunoproteasomes. The correlations between *PSMB1-10* (*PSMB8-9* especially) gene expression and ER, PR, HER-2 and Ki-67 expression levels were identified. For the first time, the linear regression method revealed a relationship reflecting the possible influence of the percentage of cells expressing estrogen and progesterone receptors, as well as Ki-67 on ChTL and CL activity of proteasomes (Figure 5).

The study of regression relationships showed that not only proteasomes can regulate the levels of ER, PR and Ki-67, but also these biological molecules probably affect the functioning of proteasomes. The identified changes in the expression of the LMP-7 subunit in tissues, characteristic to luminal cancers, can be explained by the functioning of ERα as a factor that indirectly regulates the expression of immunoproteasome subunits through the participation of microRNAs (MIR) and certain transcription factors. It is known that ER induces the expression of MIR-191 [64]. In turn, MIR-191 suppresses the expression of a number of genes, including *SOX4* in breast carcinoma [65]. The transcription factor SOX4 can influence the mRNA level of the transcription factor PU.1 [66], which directly binds and transactivates the promoters of *PSMB8* and *PSMB9* immunoproteasome genes that encode LMP-7 and LMP-2, respectively [67]. Currently, there is very little data on mechanisms that would suggest ways in which PR and Ki-67 influence the functioning of proteasomes. The Ki-67 protein has been widely used as a proliferation marker for human tumor cells, as it is necessary for cell division [68]. Ki-67 was reported to make distinct contributions to the internal organization of nucleoli and to the organization of heterochromatin around the nucleolus [69]. Thus, we speculate that these processes can modulate the expression of various genes including proteasome genes. Along these lines, it has been shown that Ki-67 knockout breast cancer cells demonstrate decreased expression of *PSMB8* and *PSMB9* [70]. Moreover, considering the important role of these molecules in the regulation of numerous processes, such as proliferation, differentiation and signaling, it can be assumed that Ki-67 might modulate the activity of ubiquitin ligases and/or proteasomes.

The importance of revealed correlations and association of a particular subtype of BC with an immunoproteasome expression pattern may be linked to several interesting implications.

Tumors were demonstrated to be highly dependent on the effective functional activity of the UPS [71]. Our previous studies have shown the ability to predict disease-free survival of BC patients by determining proteasome activity in the tumor and adjacent tissue [72]. One of the ways to regulate the proteolytic activity of proteasomes is to change their subunit composition or amount and the assortment of attached regulators [15]. Along these lines, analysis of available datasets and gene expression in BC cell lines showed that proteasome beta subunits gene expression levels were different and associated with the presence of ER, PR and Ki-67 in tumors.

Furthermore, by Western blotting we demonstrated that characteristic differences in quantity of proteasome subunits and subunits of proteasome regulators are distinctive to different subtypes of the BC. Thus, in luminal A cancer tissue, high content of 20S proteasome alpha subunits reflecting total quantity of proteasomes, as well as the proteasome regulator PA28β, were revealed. At the same time, lowest levels of immunoproteasome subunits LMP-2 and LMP-7 were observed, indicating high constitutive proteasome/immunoproteasome ratio. Luminal B subtype of the BC was characterized by highest content of LMP-7 and Rpt6 subunits, relatively low content of 20S proteasome alpha subunits and moderate amount of LMP-2 and PA28β subunits. It is known that triple-negative subtype of breast cancer is characterized by a worse prognosis compared to luminal A cancer or luminal B. Triple-negative tumors are characterized by high expression of immunoproteasome subunits, and recent reports indicate that immunoproteasome subunit expression can represent a prognostic marker of the cancer outcome [28]. Indeed, in our study these tumors demonstrated highest content of LMP-2 and moderate levels of 19S regulator subunit Rpt6, immunoproteasome subunit LMP-7 and the 20S proteasome alpha subunits, but the lowest content of PA28-β subunits.

High immunoproteasome content of triple-negative cancers might be explained as a result of efficient infiltration of tumors by tumor-infiltrating lymphocytes (TILs) [26,73]. Interestingly, bioinformatic analysis demonstrated an increase in *PSMB8-10* expression in tumors with lymph node metastases compared with tumors without metastases. One cannot exclude that, in this case, stimulated presentation of cancer antigens might facilitate migration of immune cells into the primary tumor. Along these lines, tumors with high expression of LMP-7 (encoded by *PSMB8)* were shown to have more tumor-infiltrating lymphocytes in each BC subtype [26]. At the same time, our analysis and data reported by others [28] indicated that several cell types including mammary gland cells, endocrine cells, fibroblasts andendothelial cells express considerable levels of immunoproteasome subunits (Figure 1c). If expressed by cancer cells, increased amount of constitutive and immunoproteasomes might be both a “positive” and a “negative” factor for the tumor [27,28,74,75,76]. Thus, immunoproteasomes can facilitate presentation of certain cancer antigens and by this stimulate recognition of the tumor by the immune system [27,28,77]. On the contrary, increased levels of immunoproteasomes might help cancer cells to get rid of tumor suppressor proteins, create a favorable microenvironment and deal with different types of stress [76,78,79]. Moreover, the 11Sαβ is considered to be involved in antigen presentation [13]; hence, lower content of this regulator might affect generation of certain cancer antigens [80] affecting the overall prognosis. Thus, levels of BC markers might be indicative on the levels of proteasome activity and quantity of immunoproteasomes, which eventually affects the outcome of the disease.

Obtained data also indicate that modulation of immunoproteasome activity and content might represent promising strategy for the BC management. Thus, immunoproteasome-specific inhibitors that are now being actively developed [81] might be considered for the treatment of certain types of BC [39]. Along these lines, we demonstrated that BC cell lines SKBR3 and ZR-75 characterized by highest immunoproteasome expression levels demonstrate lowest IC50 values upon treatment with the immunoproteasome inhibitor ONX-0914 (Figure 3d). This indicates that immunoproteasome inhibition may be a suitable therapeutic approach for cells expressing high levels of immunoproteasomes. On the other hand, inhibition of immunoproteasome activity in tumors might be counterproductive, at least in case of triple-negative cancer due to attenuation of anti-cancer immunity [27], indicating that, on the contrary, induction of immunoproteasomes might be effective. It is well established that immunoproteasome expression is facilitated by pro-inflammatory cytokines like IFN-γ and TNF [82]. Rather old reports already highlighted beneficial action of IFN-γ against BC [83,84]. It has been shown that elevated serum levels of IFN-γ associate with favorable disease outcome in hormone-dependent BC [85]. Moreover, IFN-γ stimulated decrease of the BC stem cell marker expression, as well as downregulating sphere formation and cancer cell invasion [86]. Though IFN-γ is a cytokine with pleotropic activity, recent reports indicate that, at least in part, its anti-cancer activity in BC is associated with the UPS modulation. IFN-γ was shown to stimulate the expression and activity of CUL ubiquitin-ligase, promoting the degradation of HER-2 and consequently diminishing growth of the tumor. Furthermore, combinations of IFN-γ with multiple HER-2-targeting drugs were shown to decrease HER-2 surface expression and suppress tumor growth [87]. Thus, it is reasonable to assume that IFN-γ effects might be in part associated with increased expression of immunoproteasomes. Concordantly, the applicability of immunoproteasome inhibitors or stimulators for the BC therapy might be deduced considering the subtype of the tumor, e.g., expression level of BC markers.

## 5. Conclusions

In various subtypes of breast cancer, regulation of ER, PR and Ki-67 expression ensures tumor progression. In recent years, a large number of studies have confirmed that the molecular indicators that determine the subtype of BC are tightly connected. The influence of the UPS on estrogen and progesterone receptors, as well as the Ki-67, was reported previously. Here, using mathematical analysis, the probable mutual influence of parameters that determine the molecular subtype of breast cancer and proteasome activity has been demonstrated. This result allows us to formulate a hypothesis about the possible mutual regulation of proteasome activity and the expression of ER, PR and Ki-67. However, confirmation of the found regression relationships is necessary and worthy of further exploration, which may have practical significance. Moreover, our data indicate that BC types differ in the expression levels of proteasome subunits and the presence of cross-regulation mechanisms between the molecules determining the subtype of breast cancer and immunoproteasomes. Currently, studies are underway on the use of proteasome modulators for the treatment of BC. Our data indicate that combined administration of these drugs and standard antitumor therapy regimens can lead to different outcomes depending on the molecular type of BC. Thus, care should be taken: The molecular subtype of BC and immunoproteasome subunit expression levels should be considered before such therapeutic modalities are applied.

## Figures and Tables

**Figure 1 cancers-17-00159-f001:**
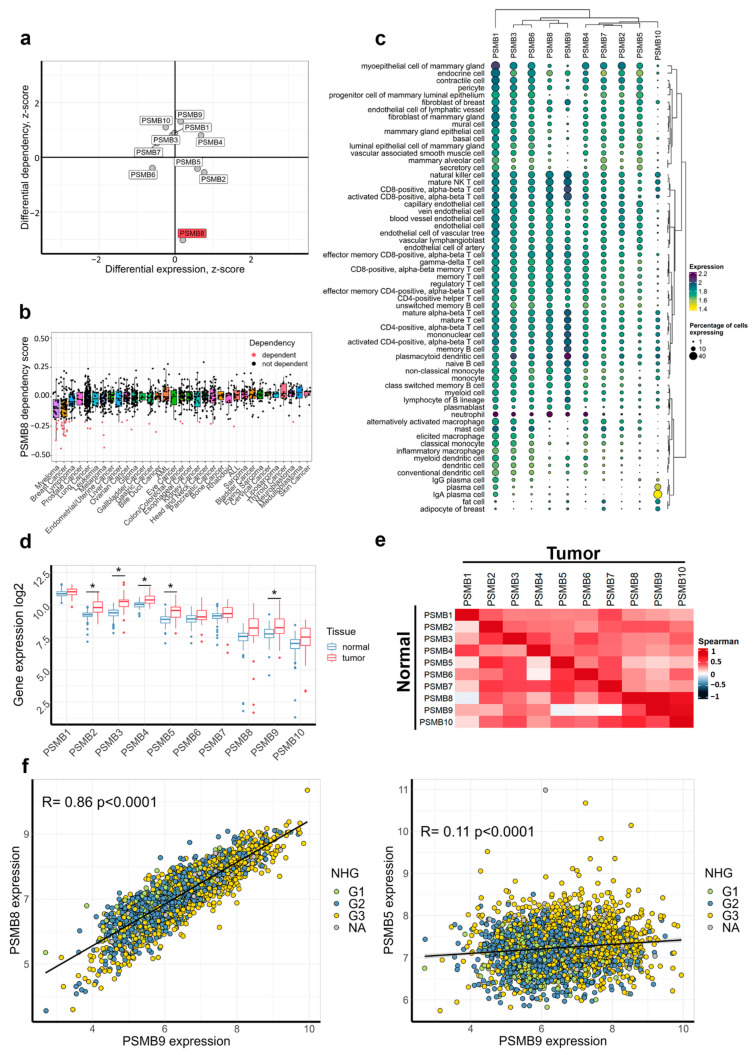
The *PSMB1-10* gene expression in breast carcinomas. (**a**) Comparison of *PSMB1-10* gene expression and gene fitness data for breast cancer cell lines. Gene expression z-scores based on analysis of 20 cancer types from 144 datasets (R2: Genomics Analysis and Visualization Platform) represent how the *PSMB1-10* gene expression in BC differs from other tumor types. Positive scores mean increased expression and negative scores - lower expression. Z-scores for gene dependency were obtained from DepMap data, and negative z-scores mean that proliferation and survival of breast cancer cells are more dependent on this gene expression than in other cell types. (**b**) The *PSMB8* dependency score across 33 cancer cell types comparing cancers from DepMap database. Negative scores mean that there are fewer cells in screens after gene knockdown/knockout. Cells with potentially reduced proliferation due to *PSMB8* downregulation are highlighted in red. (**c**) Gene expression in cell types identified in single cell data from CZ CELLxGENE Discover dataset. Color is proportional to gene expression and circle area proportional to percentage of cells expressing a gene. Data was clustered using weighted gene expression with Ward D2 algorithm. (**d**) Gene expression between malignant and histologically normal breast tissues. (**e**) Spearman correlation between *PSMB* genes in 143 normal breast tissues (lower-left half) and in 3207 breast cancer tumors (upper-right half). (**f**) Correlation between *PSMB8* and *PSMB9* (left), as well as *PSMB5* and *PSMB9* (right) gene expression in tumors. Spearman correlation was calculated, and linear regression line is shown. Color marks samples according to Nottingham histologic grading. *—*p* < 0.001 as calculated by two-sided Student’s *t*-test for each gene expression between normal and tumor tissues with Benjamini–Hochberg correction for comparing multiple genes.

**Figure 2 cancers-17-00159-f002:**
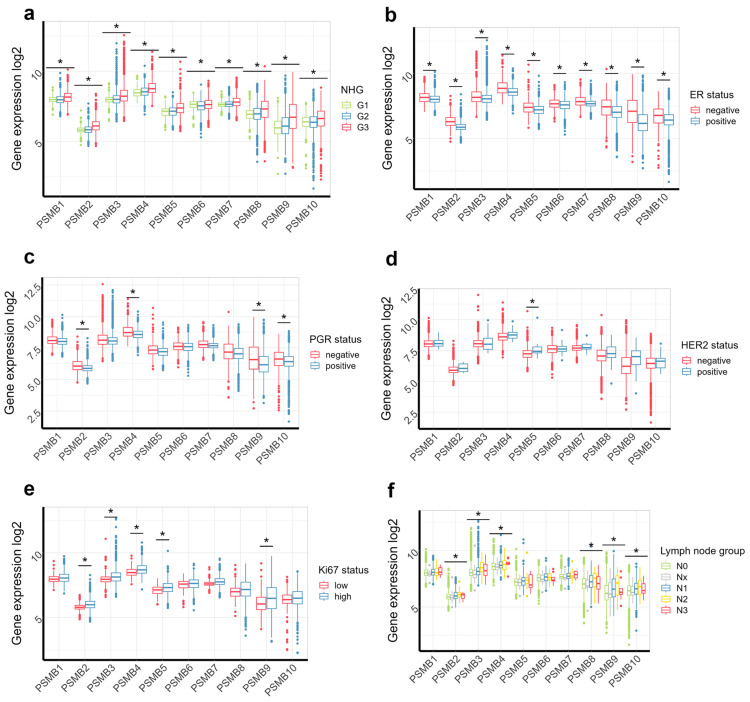
Association of *PSMB1-10* gene expression with clinical features of breast carcinomas. (**a**) The *PSMB1-10* gene expression in breast carcinomas with various grades according to Nottingham histologic grading. The *PSMB1-10* gene expression in (**b**) ER-negative vs. ER-positive, (**c**) PR-negative vs. PR-positive, (**d**) HER2-negative vs. HER2-positive, (**e**) Ki67-low vs. Ki-67 high breast carcinomas. (**f**) The *PSMB1-10* gene expression in tumors with metastasis to different amount of lymph nodes. Gene expression data was extracted from GSE202203 dataset (n = 3207). *—*p* < 0.001 as calculated by two-sided Student’s *t*-test for each gene expression between two patient groups (**b**–**e**), and one-way ANOVA was used for comparison of multiple groups (**a**,**f**) with Benjamini-Hochberg correction for comparing multiple genes.

**Figure 3 cancers-17-00159-f003:**
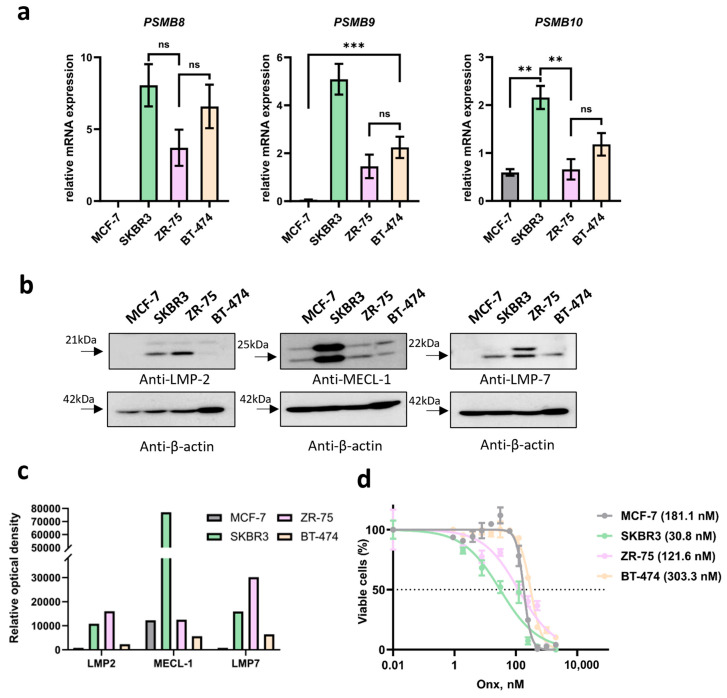
Immunoproteasome subunit expression affects sensitivity of BC cell lines of different origin to immunoproteasome inhibition. (**a**) Relative mRNA levels of *PSMB8*, *PSMB9* and *PSMB10* in MCF-7, SKBR3, ZR-75-1 and BT-474 cell lines normalized to the mean among all studied cell lines. *p*-values were calculated using Ordinary one-way ANOVA test, with significance levels denoted as follows: ns—*p* > 0.05; **—*p* < 0.01; ***—*p* < 0.001. (**b**) Western blotting of BC cell lysates with antibodies to LMP-2, MECL-1 and LMP-7. (**c**) Quantification of blots (**b**) using the ImageJ software ver.1.54i. (**d**) Viability of cells treated with ONX-0914 (Onx) for 72 h (% to DMSO-treated control) was measured using Resazurin cytotoxicity assay kit, and the half-maximal inhibitory concentrations (IC50s) were determined using nonlinear regression analysis with variable slope fitting (IC50 values represented in parentheses right side from the graph). The dotted line represents the level with 50% viability (IC50). All experiments were conducted in triplicates, and standard error of the mean (SEM) is indicated for each bar.

**Figure 4 cancers-17-00159-f004:**
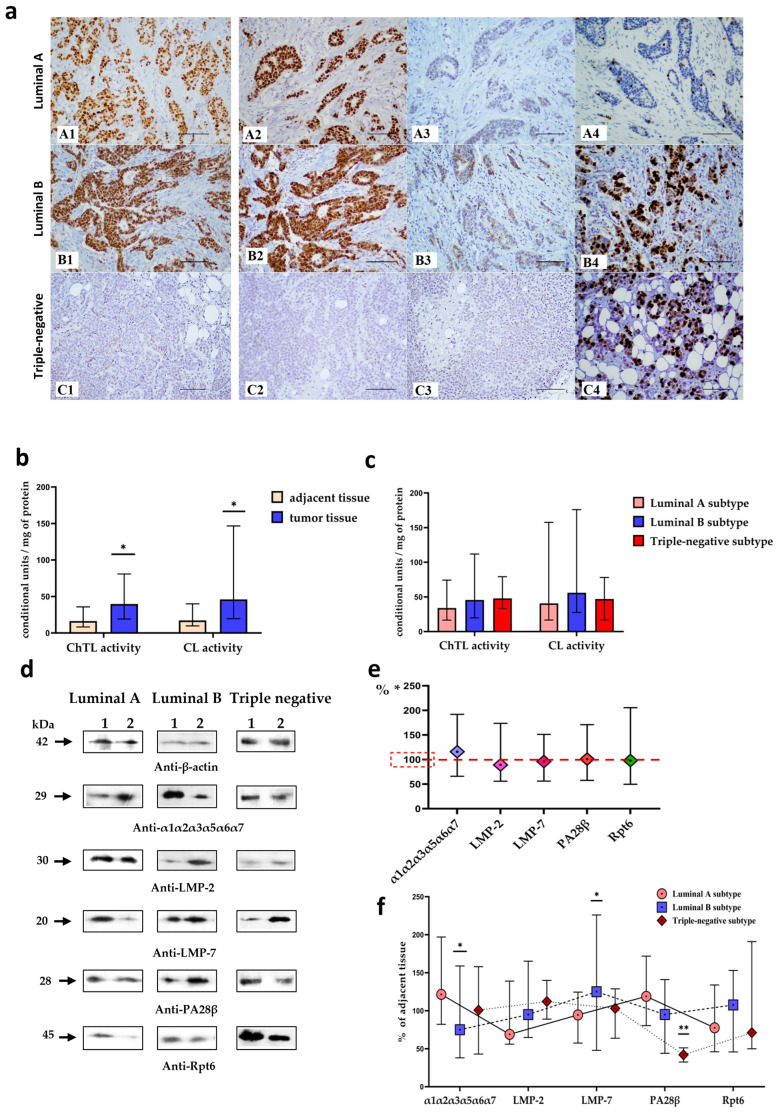
Proteasome activities and content in BC and adjacent tissues. Representative immunohistochemical staining of tumor tissues (**a**). First row (**A1**–**A4**) Invasive carcinoma of nonspecific type (Luminal A subtype): **A1**—Strong expression of ER; **A2**—Strong expression of PR; **A3**—negative expression of HER-2; **A4**—low level of Ki-67. Second row (**B1**–**B4**) Invasive carcinoma of nonspecific type (Luminal B subtype): **B1**—Strong expression of ER; **B2**—Strong expression of PR; **B3**—weak (1+) expression of HER-2; **A4**—high level of Ki-67 (>20%). Third row (**C1**–**C4**) Invasive carcinoma of nonspecific type (Triple negative subtype): **C1**—negative expression of ER; **C2**, negative expression of PR; **C3**, negative expression of HER-2; **A4**, high level of Ki-67 (>50%). Immunohistochemistry, ×200. Scale bar—100 μm. (**b**) ChTL and CL proteasome activities in BC and adjacent tissues (**c**). Proteasome activities in tissues of various molecular subtypes of BC. (**d**) The content of 20S proteasome or proteasome regulator subunits ((α1, α2, α3, α5, α6, α7), Rpt6, PA28β LMP-2, LMP-7) determined by Western blotting in BC (1) and adjacent tissues (2). Representative Western blots are shown. (**e**) Content of proteasome subunits in tumour and adjacent tissue; and in different molecular subtypes of the BC (**f**) determined via quantification of blots using the ImageJ software ver. 1.54i. Red box indicates the relative content of proteasome subunits in the adjacent tissue, which was considered as 100%; * *p* < 0.05, ** *p* < 0.01, Mann-Whitney test.

**Figure 5 cancers-17-00159-f005:**
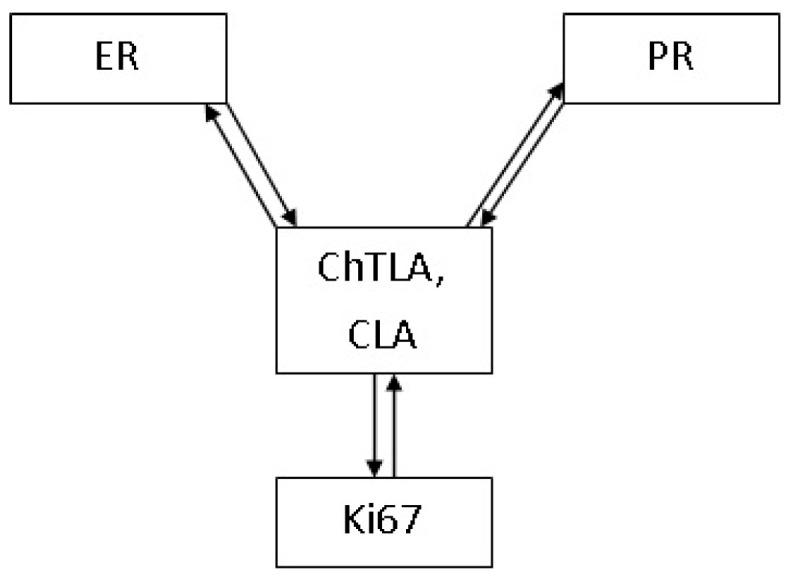
Interrelation and mutual influence of proteasome activities and indicators on the basis of which the molecular subtype of breast cancer is determined. Note: → The positive influence of one characteristic on another.

**Table 1 cancers-17-00159-t001:** Distribution of breast cancer patients depending on molecular subtype.

Molecular Markers	Breast Cancer, n—Number of Patients			
	Luminal		Triple Negative	General Cancer Group,
	A, n = 73, (46%)	B, n = 62 (39%)	n = 24 (15%)	n = 159 (100%)
Expression ER:				
Yes	73 (100%)	62 (100%)	0	135 (85%)
No	0	0	24 (100%)	24 (15%)
Expression PR:				
Yes	66 (90%)	52 (84%)	0	118 (74%)
No	7 (10%)	10 (16%)	24 (100%)	41 (26%)
Expression Ki 67				
<20%	73(100%)	0	0	73 (46%)
≥20%	0	62 (100%)	24 (100%)	86 (54%)
Expression HER-2:				
Yes	0	12 (19%)	0	12 (8%)
No	73 (100%)	50 (81%)	24 (100%)	147 (92%)

**Table 2 cancers-17-00159-t002:** Relative content of proteasome subunits in tumor tissue in different molecular subtypes of BC.

Proteasome Subunits	Molecular Subtype of Breast Cancer		
	Luminal A, n = 73	Luminal B, n = 62	Triple-Negative, n = 24
α1, α2, α3, α5, α6, α7	121.50 [82.20; 197.0]	75 [38.0; 159.0] *p*_1_ = 0.049	100.78 [42.99; 158.0]
LMP-2	69.0 [56.0; 139.0]	94.95 [64.77; 165.26]	112.11 [89.0; 140.0]
LMP-7	94.50 [57.45; 124.50]	125.0 [48.0; 226.0] *p*_1_ = 0.002	103.07 [63.85; 128.81]
PA28β	118.85 [80.41; 171.90]	95.0 [44.0; 142.01]	42 [32.57; 51.15] *p*_2_ = 0.00
Rpt6	77.60 [46.0; 133.85]	107.56 [45.80; 153.0]	71.21 [50.0; 191.0]

Note: The level of proteasome subunits in adjacent issue is taken as 100%; *p*_1_—significance of differences between luminal A and luminal B cancer; *p*_2_—significance of differences between luminal A and triple-negative breast cancer. The results in the table are presented as Me [Q1; Q3], where Me is the median, Q1 and Q3 are the lower and upper quartiles.

**Table 3 cancers-17-00159-t003:** Correlations between proteasome activities, expression of estrogen and progesterone receptors, Ki-67 and HER-2 receptor.

Pair of Variables	Spearman Rank Order Correlations			
	n	Spearman R	T(N-2)	*p*-Level
ER:PR	118	0.63	8.75	0.00
ER:Ki-67	108	−0.39	−4.47	0.00
PR:HER-2	115	−0.23	−2.55	0.01
Ki-67:HER-2	165	0.15	1.97	0.05
ChTLA:CLA	150	0.58	8.66	0.00

**Table 4 cancers-17-00159-t004:** Regression relationship reflecting the possible influence of indicators.

Dependent Variable	Predictor	Coefficient β	*p*-Value
ER, %	ChTLA	0.74	0.00
	CLA	1.89	0.00
PR,%	ChTLA	0.88	0.00
	CLA	1.88	0.00
Ki-67, %	ChTLA	1.66	0.00
	CL	3.04	0.00
ChTLA	ER, %	0.46	0.00
	PR, %	0.31	0.00
	Ki67, %	0.25	0.00
CLA	ER, %	0.15	0.00
	PR, %	0.09	0.00
	Ki67, %	0.07	0.00

## Data Availability

Data is available upon reasonable request.

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
