# Peer review of "Association of Proteasome Activity and Pool Heterogeneity with Markers Determining the Molecular Subtypes of Breast Cancer"

_cancers, 2025, doi:10.3390/cancers17010159_

Round 1
Reviewer 1 Report (New Reviewer)
Comments and Suggestions for Authors
Unfortunately, the authors have addressed my concerns just partially. The reviewer finds the improvements of the manuscript made in the revised version to be insufficient to approve the current paper for publication in the Cancers journal. The reasons are listed below:
The text is difficult to read, since the version uploaded to the editorial office contains revisions made sequentially by several persons.
Materials and Methods: The authors claim that they analyzed immunosubunits in cell lysate samples. What about the other subunits?
And why is there no mention of MECL-1 above (line 364)?
The immunoblots quality - as it was poor, it still remains, which is confirmed by the "original membranes". The blot with alpha subunits is still represented by the same one band, although there should be half a dozen of them (even if not taking into account the PTMs and/or isoforms of alpha subunits, in particular, alpha6 (PSMA1), which lead to the appearance of additional bands).
The authors answered to the reviewer the following: "By Western blotting we determined the expression on corresponding proteins (LMP7, LMP2 and MECL-1)."
Unfortunately, the revised version of the manuscript does not contain neither anti-MECL-1 nor anti-beta1-2-5 immunoblots.
The legend to Table 2 does not indicate what the values in square brackets mean.
Lines 808-809 - "amount of attached regulators"
Amount or assortment?
Lines 827-844 are quite speculative. Especially - concerning the alpha subunits. Their amount should reflect the content of the 20S proteasomes, since each core complex is composed of 14 alpha- and 14 beta-subunits. The only exception is if we are talking about free alpha subunits, also represented in the cell, but that is an another story.
Comments on the Quality of English LanguageThe reviewer finds the English language quality to be almost suitable. Just moderate editing is required
Author Response
Unfortunately, the authors have addressed my concerns just partially. The reviewer finds the improvements of the manuscript made in the revised version to be insufficient to approve the current paper for publication in the Cancers journal. The reasons are listed below:
Dear Reviewer,
Thank you for the reconsideration of our manuscript, comments and suggestions. Please find the responses below.
Comment 1:
The text is difficult to read, since the version uploaded to the editorial office contains revisions made sequentially by several persons.
Response 1:
The manuscript was significantly rewritten, many corrections were introduced and several authors contributed jointly in order to improve it. Since the submitted article should contain all the introduced changes to highlight the corrections made, we retained all the revisions. The text was additionally modified in order to increase the clarity of certain phrases.
Comment 2:
Materials and Methods: The authors claim that they analyzed immunosubunits in cell lysate samples. What about the other subunits?
Response 2:
We focused on the immunoproteasome subunits since transcriptome analysis indicated that BC types differ in terms of proteasome subunit expression pattern and together with strong PSMB8-10 co-correlation in tumors these data suggested exclusive function of immunoproteasomes in biology of the BC.
Comment 3:
And why is there no mention of MECL-1 above (line 364)?
Response 3:
The MECL-1 subunit is mentioned for the first time in the Introduction (Line 101).
Comment 4:
The immunoblots quality - as it was poor, it still remains, which is confirmed by the "original membranes". The blot with alpha subunits is still represented by the same one band, although there should be half a dozen of them (even if not taking into account the PTMs and/or isoforms of alpha subunits, in particular, alpha6 (PSMA1), which lead to the appearance of additional bands).
Response 4:
Indeed, staining with the antibodies to the alpha subunits of the proteasome should give at least 6 bands. However, several alpha subunits of the proteasome have close MW, e.g. 25.7; 26.4; 27.3; 27.8; 28.3; 29.5; 29.8 kDa. Thus, several proteins can migrate very close in the gel and one can expect “fusion” of several bands in one in Western blot, therefore reducing the quantity of visible bands. Our experience indicates that from 3 to 4 subunits are seen on the blot when such antibodies are used. However, in certain cases PTMs and mutations can affect not only the position of the protein in the gel, but also the efficacy of the antibody binding. This is particularly relevant since the used antibodies represent a combination of monoclonal antibodies to different subunits. One cannot exclude also a different stability of monoclonal antibodies within the mixture. Altogether these factors can reduce the amount of bands seen after the development of the blot.
Comment 5:
The authors answered to the reviewer the following: "By Western blotting we determined the expression on corresponding proteins (LMP7, LMP2 and MECL-1)."
Response 5:
Please consider our previous response as a whole.
“We performed additional experiments using antibodies from another supplier. We have tested the expression levels of immunoproteasome subunits in four different BC cell lines, which were derived from patients with a particular BC subtype. We analyzed the expression of PSMB8,9,10 genes by qPCR. By Western blotting we determined the expression on corresponding proteins (LMP7,LMP2 and MECL-1). Moreover we estimated the sensitivity of those cell lines to the immunoproteasome inhibitor ONX-0914.”
We indicated that by Western blotting we tested the expression of proteasome subunits (LMP2, MECL-1 and LMP7) in four different BC cell lines (Figure 3). Moreover, we used antibodies from another supplier, as was suggested by the reviewer.
Comment 6:
Unfortunately, the revised version of the manuscript does not contain neither anti-MECL-1 nor anti-beta1-2-5 immunoblots.
Response 6:
For the Western blot with anti-MECL-1 antibodies, please refer to Figure 3. Blots using antibodies to β1, 2, 5 were not performed since we were focused on the expression of the immunoproteasome subunits for the reasons indicated above.
Unfortunately, the tumor lysates were entirely used previously and it is quite difficult to gather new samples, moreover it will lead to a necessity to reperform all the experiments from the second part of the manuscript.
Comment 7:
The legend to Table 2 does not indicate what the values in square brackets mean.
Response 7:
Corrected. We indicated: The results in the table are presented as Me [Q1;Q3], where Me is the median, Q1 and Q3 are the lower and upper quartiles.
Comment 8:
Lines 808-809 - "amount of attached regulators"
Amount or assortment?
Response 8:
Corrected for “amount and assortment”. The use of word “amount” is based on the fact that association of proteasomes with regulators is reversible and proteasomes can dissociate or reassociate with regulators. Thus, not only the assortment, but also the amount of regulators attached to proteasomes at a given time point affects the proteasome activity.
Comment 9:
Lines 827-844 are quite speculative. Especially - concerning the alpha subunits. Their amount should reflect the content of the 20S proteasomes, since each core complex is composed of 14 alpha- and 14 beta-subunits. The only exception is if we are talking about free alpha subunits, also represented in the cell, but that is an another story.
Response 9:
We agree with the reviewer that the amount of 20S alpha subunits generally reflects the overall quantity of proteasomes. Our data highlights differences in the amount of alpha and immune beta proteasome subunits, thus indicating that the ratio of constitutive/immune proteasomes might be unequal between different BC subtypes. We introduced relevant corrections into the text Lines 690-693.
Reviewer 2 Report (New Reviewer)
Comments and Suggestions for Authors
The study conducted by Kondakova and colleagues investigated potential correlations in the expression of various proteasome subunits, which are responsible for the constitutive and functional modification of this multicatalytic complex in different well-characterized breast cancer subtypes, identified through specific markers such as ER, PGR, HER-2, and Ki-67. The findings indicate that the dynamics of proteasome assembly may vary across tumor types. Additionally, the expression of PSMB8-10 subunits, associated with the immunoproteasome configuration, showed a strong correlation in these tumors. These results are particularly relevant since proteasome inhibitors are available as a therapeutic option, and the response to these treatments may differ among tumor types due to the observed heterogeneity in proteasome composition. The article's introduction and discussion sections effectively highlighted the main research points, providing in-depth insights.The data obtained were supported by bioinformatics analyses of patient and single-cell databases, as well as experiments with tumor cell lines, with all tumor types and groups well-defined and characterized using classical markers.
Minor:
-
Lines 25, 26, and 27: Rewritten into a single paragraph for better flow and to avoid repetition.
-
Mention of PSMB1-10 in the Introduction. It is important to correlate this term in the text to enhance the reader's understanding.
-
Line 271: Use SDS-PAGE instead of PAAG to align with standard terminology or Add a clarification about the meaning of PAAG (Polyacrylamide Gel Electrophoresis) for context.
-
Line 188: Explain the function of ONX-0914 in this section, not just in the results. For example, mention that ONX-0914 is a selective immunoproteasome inhibitor
Author Response
The study conducted by Kondakova and colleagues investigated potential correlations in the expression of various proteasome subunits, which are responsible for the constitutive and functional modification of this multicatalytic complex in different well-characterized breast cancer subtypes, identified through specific markers such as ER, PGR, HER-2, and Ki-67. The findings indicate that the dynamics of proteasome assembly may vary across tumor types. Additionally, the expression of PSMB8-10 subunits, associated with the immunoproteasome configuration, showed a strong correlation in these tumors. These results are particularly relevant since proteasome inhibitors are available as a therapeutic option, and the response to these treatments may differ among tumor types due to the observed heterogeneity in proteasome composition. The article's introduction and discussion sections effectively highlighted the main research points, providing in-depth insights.The data obtained were supported by bioinformatics analyses of patient and single-cell databases, as well as experiments with tumor cell lines, with all tumor types and groups well-defined and characterized using classical markers.
Dear Reviewer,
Thank You very much for reconsideration and evaluation of our manuscript, comments and suggestions.
Please find the responses below
Comment 1:
Minor:
Lines 25, 26, and 27: Rewritten into a single paragraph for better flow and to avoid repetition.
Response 1:
The Simple Summary was included in accordance with the Cancers template as an obligatory part of the manuscript. Therefore we cannot merge the Simple Summary and the Abstract.
Comment 2:
Mention of PSMB1-10 in the Introduction. It is important to correlate this term in the text to enhance the reader's understanding.
Response 2:
Corrected.
Comment 3:
Line 271: Use SDS-PAGE instead of PAAG to align with standard terminology or Add a clarification about the meaning of PAAG (Polyacrylamide Gel Electrophoresis) for context.
Response 3:
Corrected.
Comment 4:
Line 188: Explain the function of ONX-0914 in this section, not just in the results. For example, mention that ONX-0914 is a selective immunoproteasome inhibitor
Response 4:
Corrected.
Round 2
Reviewer 1 Report (New Reviewer)
Comments and Suggestions for Authors
The authors have addressed most of my concerns. The reviewer finds the revised version of the manuscript much improved in term of data presentation and organization and approves it for the publication in the Cancers journal. However, the reviewer can't agree with authors' response 4 (Fig.4d, anti-alpha subunits immunoblot): at least several distinct bands should appear on the blot instead of one band in Fig.4d of the present manuscript.
Comments on the Quality of English LanguageThe reviewer finds the English language quality to be suitable. Just several typos were detected.
Author Response
Comment:
The authors have addressed most of my concerns. The reviewer finds the revised version of the manuscript much improved in term of data presentation and organization and approves it for the publication in the Cancers journal. However, the reviewer can't agree with authors' response 4 (Fig.4d, anti-alpha subunits immunoblot): at least several distinct bands should appear on the blot instead of one band in Fig.4d of the present manuscript.
Response:
Thank you. We agree and understand the concerns of the Reviewer. We introduced a relevant correction into the text (Lines 556-560). We indicated: “Notably, a single protein band was frequently observed in Western blotting with antibodies to α1,α2,α3,α5,α6,α7. This effect might be a result of overlap of signals from proteasome subunits with close MW, post-translational modifications of proteasome subunits or different avidity of monoclonal antibodies directed to particular subunits.”
Indeed, the detection of 20S alpha subunits using anti-alpha1,2,3,5,6,7 antibodies could come out with different amount of bands in WB. First, some 20S proteasome subunits have a very close molecular weight. In this regards it is quite difficult to well separate protein subunits even if using both sucrose gradient centrifugation and gradient gel electrophoresis. Pure proteasome preparations analyzed by centrifugation in sucrose gradient and gradient PAGE reveal only 6-7 discrete bands instead of 14 protein bands (Figure 1a, PMID: 39519262). This quantity accounts for both alpha and beta proteasome subunits. As we indicated previously, 20S alpha subunits have a MW of 25.7; 26.4; 27.3; 27.8; 28.3; 29.5; 29.8 kDa. Therefore, when we use antibodies to 6 out of the 7 alpha subunits we normally observe less than 6 discrete bands due to the close migration (overlap) of subunits in the gel. This effect is more evident when a gel with a fixed per cent of acrylamide is used instead of a gradient gel. Except close MW of particular subunits there are several reasons why fewer amount of bands could be seen after the WB:
- Next, the concentration of cellular proteins in lysates is critical for the proteasome detection. We use a mixture of antibodies, but antibodies can demonstrate different avidity, which affects the binding to a particular proteasome subunit. Therefore, fluctuations in the amount of lysate or concentration of antibody in the reaction might unequally affect the efficacy of proteasome subunit detection in WB. This means that intensity of some bands could be much higher than of others.
- Mutations and PTMs of proteasome subunits can affect not only the position of the protein in the gel, but also the efficacy of the antibody binding. As we indicated in the previous response, this is particularly relevant since used antibodies represent a combination of monoclonal antibodies directed to different subunits. Therefore, when proteasomes in lysates obtained from different types of cells are analyzed, different outcomes might be expected. Please refer the image from the manufacturer’s web site https://www.enzo.com/product/proteasome-20s-%CE%B11-2-3-5-6-7-subunits-monoclonal-antibody-mcp231/ the proteasome subunit profile in lysates of HeLa and Jurkat cells is different.
- Another reason might be associated with the exposure time and the sensitivity of ECL detection system.
- Finally, please refer to images (results) from published papers where anti-20S α1,2,3,5,6,7 was used for the proteasome subunits detection using ECL WB. From 1 to 4 proteasome subunits could be seen, please explore Fig. 2B (PMID: 28088524), Fig. 1A (PMID:25634956), Fig. 6AB(PMID:23536702), Fig. 3B(PMID:32306217), Fig. 7C (PMID:21998752).
This manuscript is a resubmission of an earlier submission. The following is a list of the peer review reports and author responses from that submission.
Round 1
Reviewer 1 Report
Comments and Suggestions for Authors
This is a descriptive study that compares proteasome expression in different subtypes of breast cancer. It’s significance for the understanding of mechanisms, and for the improvement of diagnostic and treatment of breast cancer is not clear.
The lethal flaw of this study is that authors used Z-LLG-amc to measure caspase-like activity of the proteasome. Z-LLG-amc is not a substrate of the caspase-like site. It is much more likely to be cleaved by neutrophil elastase.
In all their assay, the authors measured proteasome activity in crude extracts without determining the background due to the cleavage by non-proteasomal proteinases.
Comparison of activity between tumor and adjacent normal tissues does not make sense without knowing that the normal tissue is epithelial tissue. Malignant breast cells are epithelial. Mammary glands contain a lot of fat, and if proteasome activity in fat tissues is lower, activity in the adjacent tissues will be lower. The background in proteasome activity assays may be different between epithelial and fat tissues. Activity will also be affected by immune infiltrates, and increased LMP7 activity in luminal B samples may reflect just that.
Blots on Fig. 3d are meaningless because you cannot compare band intensity between different membranes without a reference sample present on all of them.
Minor point:
Decimals in English are separated by periods, not commas.
HER2-positive tumors are not the worse in terms of prognosis, triple negative are.
Reviewer 2 Report
Comments and Suggestions for Authors
the manuscript titled as "Association of the Ubiquitin-Proteasome System with Markers Determining the Molecular Subtypes of Breast Cancer" concludes that different types of BC demonstrate different patterns of proteasome genes and proteins expression. Unexpected links between the proteasome, hormone receptors and Ki67 indicate mutual influence and crosstalk mechanisms, which could be used to facilitate development of novel BC subtype-specific drugs. The evidence provided by this manuscript is too weak to support their hypothesis. In addition, the novelty of the study is unapparent.
1.The manuscript does not clearly highlight what new insights or contributions it brings to the existing body of research on the ubiquitin-proteasome system (UPS) and breast cancer. especially the clinical significance held by this manuscript.
2.The manuscript contains some grammatical errors and awkward phrasings that could be improved for better readability. A thorough proofreading and editing process would be beneficial. For instance, consistent use of terms like "adjacent tissue" vs. "unaltered tissue" should be maintained throughout the manuscript to avoid confusion. The phrase "proteasome genes and proteins expression" should be corrected to "proteasome gene and protein expression."
3. The manuscript mentions the use of t-tests and ANOVA but does not specify which specific tests were used for each analysis. Additionally, the justification for choosing these tests over others should also be provided.
4. The study's sample size of 159 patients is relatively small for drawing broad conclusions about breast cancer subtypes. Furthermore, the distribution of patients across the different subtypes (e.g., luminal A, luminal B, and triple-negative) should be more balanced to ensure robust comparisons.
5. The discussion section does not adequately address the limitations of the study, such as potential biases or confounding factors. It also lacks a detailed comparison with previous studies, which would help contextualize the findings and highlight their significance.
6. Some references appear outdated, and there may be more recent studies that should be included to provide a more current context for the research. Additionally, some references are not directly relevant to the points being made in the manuscript.
7.Authors concluded that “To understand the mechanisms of breast cancer (BC) heterogeneity, it is important to study possible associations of indicators used to determine the molecular subtype of the disease (ER, PR, Ki67) with components of crucial cellular regulatory systems, including the ubiuitin-proteasome system (UPS)”. However, it seems that the relationships among ER, PR, Ki67 and UPS had not been clearly revealed in this manuscript.
Comments on the Quality of English LanguageThe manuscript contains some grammatical errors and awkward phrasings that could be improved for better readability. A thorough proofreading and editing process would be beneficial. For instance, consistent use of terms like "adjacent tissue" vs. "unaltered tissue" should be maintained throughout the manuscript to avoid confusion. The phrase "proteasome genes and proteins expression" should be corrected to "proteasome gene and protein expression."